# Implicit Finite-Horizon Approximation and Efficient Optimal Algorithms for Stochastic Shortest Path

**Liyu Chen**
University of Southern California
liyuc@usc.edu

**Mehdi Jafarnia-Jahromi**
University of Southern California
mjafarni@usc.edu

**Rahul Jain**
University of Southern California
rahul.jain@usc.edu

**Haipeng Luo**
University of Southern California
haipengl@usc.edu

## Abstract

We introduce a generic template for developing regret minimization algorithms in the Stochastic Shortest Path (SSP) model, which achieves minimax optimal regret as long as certain properties are ensured. The key of our analysis is a new technique called implicit finite-horizon approximation, which approximates the SSP model by a finite-horizon counterpart *only in the analysis* without explicit implementation. Using this template, we develop two new algorithms: the first one is model-free (the first in the literature to our knowledge) and minimax optimal under strictly positive costs; the second one is model-based and minimax optimal even with zero-cost state-action pairs, matching the best existing result from [Tarbouriech et al., 2021b]. Importantly, both algorithms admit highly sparse updates, making them computationally more efficient than all existing algorithms. Moreover, both can be made completely parameter-free.

## 1 Introduction

We study the Stochastic Shortest Path (SSP) model, where an agent aims to reach a goal state with minimum cost in a stochastic environment. SSP is well-suited for modeling many real-world applications, such as robotic manipulation, car navigation, and others. Although it is widely studied empirically (e.g., [Andrychowicz et al., 2017, Nasiriany et al., 2019]) and in optimal control theory (e.g., [Bertsekas and Tsitsiklis, 1991, Bertsekas and Yu, 2013]), it has received less attention under the regret minimization setting where a learner needs to learn the environment and improve her policy on-the-fly through repeated interaction. Specifically, the problem proceeds in $K$ episodes. In each episode, the learner starts at a fixed initial state, sequentially takes action, suffers some cost, and transits to the next state, until reaching a predefined goal state. The performance of the learner is measured by her regret, which is the difference between her total costs and that of the best policy.

Tarbouriech et al. [2020a] develop the first regret minimization algorithm for SSP with a regret bound of $\tilde{\mathcal{O}}(D^{3/2}S\sqrt{AK/c_{\min}})$, where $D$ is the diameter, $S$ is the number of states, $A$ is the number of actions, and $c_{\min}$ is the minimum cost among all state-action pairs. Cohen et al. [2020] improve over their results and give a near optimal regret bound of $\tilde{\mathcal{O}}(B_\star S\sqrt{AK})$, where $B_\star \leq D$ is the largest expected cost of the optimal policy starting from any state. Even more recently, Cohen et al. [2021] achieve minimax regret of $\tilde{\mathcal{O}}(B_\star\sqrt{SAK})$ through a finite-horizon reduction technique, and concurrently Tarbouriech et al. [2021b] also propose minimax optimal and parameter-free algorithms. Notably, all existing algorithms are model-based with space complexity $\Omega(S^2A)$. Moreover, they all

35th Conference on Neural Information Processing Systems (NeurIPS 2021).

update the learner's policy through full-planning (a term taken from [Efroni et al., 2019]), incurring a relatively high time complexity.

In this work, we further advance the state-of-the-art by proposing a generic template for regret minimization algorithms in SSP (Algorithm 1), which achieves minimax optimal regret as long as some properties are ensured. By instantiating our template differently, we make the following two key algorithmic contributions:

- In Section 4, we develop the *first model-free* SSP algorithm called LCB-ADVANTAGE-SSP (Algorithm 2). Similar to most model-free reinforcement learning algorithms, LCB-ADVANTAGE-SSP does not estimate the transition directly, enjoys a space complexity of $\tilde{\mathcal{O}}(SA)$, and also takes only $\mathcal{O}(1)$ time to update certain statistics in each step, making it a highly efficient algorithm. It achieves a regret bound of $\tilde{\mathcal{O}}(B_\star\sqrt{SAK} + B_\star^5 S^2 A/c_{\min}^4)$, which is minimax optimal when $c_{\min} > 0$. Moreover, it can be made parameter-free without worsening the regret bound.

- In Section 5, we develop another simple model-based algorithm called SVI-SSP (Algorithm 3), which achieves minimax regret $\tilde{\mathcal{O}}(B_\star\sqrt{SAK} + B_\star S^2 A)$ even when $c_{\min} = 0$, matching the best existing result by Tarbouriech et al. [2021b].[1] Notably, compared to their algorithm (as well as other model-based algorithms), SVI-SSP is computationally much more efficient since it updates each state-action pair only logarithmically many times, and each update only performs *one-step planning* (again, a term taken from [Efroni et al., 2019]) as opposed to full-planning (such as value iteration or extended value iteration); see more concrete time complexity comparisons in Section 5. SVI-SSP can also be made parameter-free following the idea of [Tarbouriech et al., 2021b].

We include a summary of regret bounds of all existing SSP algorithms as well as more complexity comparisons in Appendix A.

**Techniques**  Our main technical contribution is a new analysis framework called *implicit finite-horizon approximation* (Section 3), which is the key to analyze algorithms developed from our template. The high level idea is to approximate an SSP instance by a finite-horizon counterpart. However, the approximation *only happens in the analysis*, a key difference compared to [Chen et al., 2021, Chen and Luo, 2021, Cohen et al., 2021] that explicitly implement such an approximation in their algorithms. As a result, our method not only avoids blowing up the space complexity by a factor of the horizon, but also allows one to derive a horizon-free regret bound (more explanation to follow).

In order to achieve the minimax optimal regret, our model-free algorithm LCB-ADVANTAGE-SSP uses a key variance reduction idea via a reference-advantage decomposition by [Zhang et al., 2020b]. However, crucial distinctions exist. For example, we update the reference value function more frequently instead of only one time, which helps reduce the sample complexity and improve the lower-order term in the regret bound. We also maintain an empirical upper bound on the value function in a doubling manner, which is the key to eventually make the algorithm parameter-free. On the other hand, for our model-based algorithm SVI-SSP, we adopt a special Bernstein-style bonus term and bound the learner's total variance via recursion, taking inspiration from [Tarbouriech et al., 2021b, Zhang et al., 2020a].

**Empirical Evaluation**  We support our theoretical findings with experiments in Appendix H. Our model-free algorithm demonstrates a better convergence rate compared to vanilla Q learning with naive $\epsilon$-greedy exploration. Our model-based algorithm has competitive performance compared to other model-based algorithms, while spending the least amount of time in updates.

**Related Work**  For a detailed comparison of existing results for the same problem, we refer the readers to [Tarbouriech et al., 2021b, Table 1] as well as our Table 1. There are also several works [Rosenberg and Mansour, 2020, Chen et al., 2021, Chen and Luo, 2021] that consider the even more challenging SSP setting where the cost function is decided by an adversary and can change over time. Apart from regret minimization, Tarbouriech et al. [2021a] study the sample complexity of SSP with a generative model; Lim and Auer [2012] and Tarbouriech et al. [2020b] investigate exploration problems involving multiple goal states (multi-goal SSP).

---

[1]Depending on the available prior knowledge, the final bounds achieved by SVI-SSP are slightly different, but they all match that of EB-SSP. See [Tarbouriech et al., 2021b, Table 1] for more details.

The special case of SSP with a fixed horizon has been studied extensively, for both stochastic costs (e.g., [Azar et al., 2017, Jin et al., 2018, Efroni et al., 2019, Zanette and Brunskill, 2019, Zhang et al., 2020a]) and adversarial costs (e.g., [Neu et al., 2012, Zimin and Neu, 2013, Rosenberg and Mansour, 2019, Jin et al., 2020]). Importantly, recent works [Wang et al., 2020, Zhang et al., 2020a] find that when the cost for each episode is at most a constant, it is in fact possible to obtain a regret bound with only logarithmic dependency on the horizon. Tarbouriech et al. [2021b] generalize this concept to SSP and define horizon-free regret as a bound with only logarithmic dependence on the expected hitting time of the optimal policy starting from any state (which is bounded by $B_\star/c_{\min}$). They also propose the first algorithm with horizon-free regret for SSP, which is important for arguing minimax optimality even when $c_{\min} = 0$. Notably, our model-based algorithm SVI-SSP also achieves horizon-free regret (but the model-free one does not).

## 2 Preliminaries

An SSP instance is defined by a Markov Decision Process (MDP) $M = (\mathcal{S}, \mathcal{A}, s_{\text{init}}, g, c, P)$, where $\mathcal{S}$ is the state space, $\mathcal{A}$ is the action space, $s_{\text{init}} \in \mathcal{S}$ is the initial state, and $g \notin \mathcal{S}$ is the goal state. When taking action $a$ in state $s$, the learner suffers a cost drawn in an i.i.d manner from an unknown distribution with mean $c(s, a) \in [0, 1]$ and support $[c_{\min}, 1]$ ($c_{\min} \geq 0$), and then transits to the next state $s' \in \mathcal{S}^+ = \mathcal{S} \cup \{g\}$ with probability $P_{s,a}(s')$. We assume that the transition $P$ and the cost mean $c$ are unknown to the learner, while all other parameters are known.

The learning process goes as follows: the learner interacts with the environment for $K$ episodes. In the $k$-th episode, the learner starts in initial state $s_{\text{init}}$, sequentially takes an action, suffers a cost, and transits to the next state until reaching the goal state $g$. More formally, at the $i$-th step of the $k$-th episode, the learner observes the current state $s_i^k$ (with $s_1^k = s_{\text{init}}$), takes action $a_i^k$, suffers a cost $c_i^k$, and transits to the next state $s_{i+1}^k \sim P_{s_i^k, a_i^k}$. An episode ends when the current state is $g$, and we define the length of episode $k$ as $I_k$, such that $s_{I_k+1}^k = g$.

**Learning Objective**  At a high level, the learner's goal is to reach the goal with a small total cost. To this end, we focus on *proper policies* — a (stationary and deterministic) policy $\pi : \mathcal{S} \to \mathcal{A}$ is a mapping that assigns an action $\pi(s)$ to each state $s \in \mathcal{S}$, and it is proper if the goal is reached with probability 1 when following $\pi$ (that is, taking action $\pi(s)$ whenever in state $s$). Given a proper policy $\pi$, one can define the cost-to-go function $V^\pi : \mathcal{S} \to [0, \infty)$ as $V^\pi(s) = \mathbb{E}\left[ \sum_{i=1}^I c_i \middle| P, \pi, s_1 = s \right]$, where the expectation is with respect to the randomness of the cost $c_i$ incurred at state-action pair $(s_i, \pi(s_i))$, next state $s_{i+1} \sim P_{s_i, \pi(s_i)}$, and the number of steps $I$ before reaching $g$. The optimal proper policy $\pi^\star$ is then defined as a policy such that $V^{\pi^\star}(s) = \min_{\pi \in \Pi} V^\pi(s)$ for all $s \in \mathcal{S}$, where $\Pi$ is the set of all proper policies assumed to be nonempty. The formal objective of the learner is then to minimize her regret against $\pi^\star$, the difference between her total cost and that of the optimal proper policy, defined as

$$R_K = \sum_{k=1}^K \sum_{i=1}^{I_k} c_i^k - K \cdot V^\star(s_{\text{init}}),$$

where we use $V^\star$ as a shorthand for $V^{\pi^\star}$. The minimax optimal regret is known to be $\tilde{\mathcal{O}}(B_\star \sqrt{SAK})$, where $B_\star = \max_{s \in \mathcal{S}} V^\star(s)$, and $S = |\mathcal{S}^+|$ and $A = |\mathcal{A}|$ are the numbers of states (including the goal state) and actions respectively [Cohen et al., 2020].

**Bellman Optimality Equation**  For a proper policy $\pi$, the corresponding action-value function $Q^\pi : \mathcal{S} \times \mathcal{A} \to [0, \infty)$ is defined as $Q^\pi(s, a) = c(s, a) + \mathbb{E}_{s' \sim P_{s,a}}[V^\pi(s')]$. Similarly, we use $Q^\star$ as a shorthand for $Q^{\pi^\star}$. it is known that $\pi^\star$ satisfies the Bellman optimality equation: $V^\star(s) = \min_{a \in \mathcal{A}} Q^\star(s, a)$ for all $s \in \mathcal{S}$ [Bertsekas and Tsitsiklis, 1991].

**Assumption on $c_{\min}$**  Similar to many previous works, our analysis requires $c_{\min}$ being known and strictly positive. When $c_{\min}$ is unknown or known to be 0, a simple workaround is to solve a modified SSP instance with all observed costs clipped to $\epsilon$ if they are below some $\epsilon > 0$, so that $c_{\min} = \epsilon > 0$. Then the regret in this modified SSP is similar to that in the original SSP up to an additive term of order $\mathcal{O}(\epsilon K)$ [Tarbouriech et al., 2020a]. Therefore, throughout the paper we assume that $c_{\min}$ is known and strictly positive unless explicitly stated otherwise.

---
**Algorithm 1** A General Algorithmic Template for SSP
---
**Initialize:** $t \leftarrow 0$, $s_1 \leftarrow s_{\text{init}}$, $Q(s,a) \leftarrow 0$ for all $(s,a) \in \mathcal{S} \times \mathcal{A}$.
**for** $k = 1, \ldots, K$ **do**
   | **repeat**
1   |   | Increment time step $t \xleftarrow{+} 1$.
2   |   | Take action $a_t = \operatorname{argmin}_a Q(s_t, a)$, suffer cost $c_t$, transit to and observe $s'_t$.
3   |   | Update $Q$ (so that it satisfies Property 1 and Property 2).
4   |   | **if** $s'_t \neq g$ **then** $s_{t+1} \leftarrow s'_t$; **else** $s_{t+1} \leftarrow s_{\text{init}}$, **break**.

   Record $T \leftarrow t$ (that is, the total number of steps).
---

**Other Notations** For simplicity, we use $C_K = \sum_{k=1}^{K} \sum_{i=1}^{I_k} c_i^k$ in the analysis to denote the total costs suffered by the learner over $K$ episodes. For a function $X : \mathcal{S}^+ \to \mathbb{R}$ and a distribution $P$ over $\mathcal{S}^+$, denote by $PX = \mathbb{E}_{S \sim P}[X(S)]$, $PX^2 = \mathbb{E}_{S \sim P}[X(S)^2]$, and $\mathbb{V}(P, X) = \operatorname{VAR}_{S \sim P}[X(S)]$ the expectation, second moment, and variance of $X(S)$ respectively where $S$ is drawn from $P$. For a scalar $x$, define $(x)_+ = \max\{x, 0\}$, and denote by $\lceil x \rceil_2 = 2^{\lceil \log_2 x \rceil}$ and $\lfloor x \rfloor_2 = 2^{\lfloor \log_2 x \rfloor}$ the closest power of two upper and lower bounding $x$ respectively. For an integer $m$, $[m]$ denotes the set $\{1, \ldots, m\}$. In pseudocode, $x \xleftarrow{+} y$ is a shorthand for the increment operation $x \leftarrow x + y$.

## 3 Implicit Finite-Horizon Approximation

In this section, we introduce our main analytical technique, that is, implicitly approximating the SSP problem with a finite-horizon counterpart. We start with a general template of our algorithms shown in Algorithm 1. For notational convenience, we concatenate state-action-cost trajectories of all episodes as one single sequence $(s_t, a_t, c_t)$ for $t = 1, 2, \ldots, T$, where $s_t \in \mathcal{S}$ is one of the non-goal state, $a_t \in \mathcal{A}$ is the action taken at $s_t$, and $c_t$ is the resulting cost incurred by the learner. Note that the goal state $g$ is never included in this sequence (since no action is taken there), and we also use the notation $s'_t \in \mathcal{S}^+$ to denote the next-state following $(s_t, a_t)$, so that $s_{t+1}$ is simply $s'_t$ unless $s'_t = g$ (in which case $s_{t+1}$ is reset to the initial state $s_{\text{init}}$); see Line 4.

The template follows a rather standard idea for many reinforcement learning algorithms: maintain an (optimistic) estimate $Q$ of the optimal action-value function $Q^\star$, and act greedily by taking the action with the smallest estimate: $a_t = \operatorname{argmin}_a Q(s_t, a)$; see Line 2. The key of the analysis is often to bound the estimation error $Q^\star(s_t, a_t) - Q(s_t, a_t)$, which is relatively straightforward in a discounted setting (where the discount factor controls the growth of the error) or a finite-horizon setting (where the error vanishes after a fixed number of steps), but becomes highly non-trivial for SSP due to the lack of similar structures.

A natural idea is to explicitly solve a discounted problem or a finite-horizon problem that approximates the original SSP well enough. Unfortunately, both approaches are problematic: approximating an undiscounted MDP by a discounted one often leads to suboptimal regret [Wei et al., 2020]; on the other hand, while explicitly approximating SSP with a finite-horizon problem can lead to optimal regret [Chen et al., 2021, Cohen et al., 2021], it greatly increases the space complexity of the algorithm, and also produces non-stationary policies, which is unnatural and introduces unnecessary complexity since the optimal policy in SSP is stationary.

Therefore, we propose to approximate the original SSP instance $M$ with a finite-horizon counterpart $\widetilde{M}$ implicitly (that is, only in the analysis). We defer the formal definition of $\widetilde{M}$ to Appendix C, which is similar to those in [Chen et al., 2021, Cohen et al., 2021] and corresponds to interacting with the original SSP for $H$ steps (for some integer $H$) and then teleporting to the goal. All we need in the analysis are the optimal value function $V_h^\star$ and optimal action-value function $Q_h^\star$ of $\widetilde{M}$ for each step $h \in [H]$, which can be defined recursively without resorting to the definition of $\widetilde{M}$:

$$Q_h^\star(s,a) = c(s,a) + P_{s,a} V_{h-1}^\star, \qquad V_h^\star(s) = \min_a Q_h^\star(s,a), \qquad (1)$$

with $Q_0^\star(s,a) = 0$ for all $(s,a)$.[2] Intuitively, $Q_H^\star$ approximates $Q^\star$ well when $H$ is large enough. This is formally summarized in the lemma below, whose proof is similar to prior works (see Appendix C).

**Lemma 1.** *For any value of $H$, $Q_H^\star(s,a) \leq Q^\star(s,a)$ holds for all $(s,a)$. For any $\beta \in (0,1)$, if $H \geq \frac{4B_\star}{c_{\min}} \ln(2/\beta) + 1$, then $Q^\star(s,a) \leq Q_H^\star(s,a) + B_\star\beta$ holds for all $(s,a)$.*

In the remaining discussion, we fix a particular value of $H$. To carry out the regret analysis, we now specify two general requirements of the estimate $Q$. Let $Q_t$ be the value of $Q$ at the beginning of time step $t$ (that is, the value used in finding $a_t$). Then $Q_t$ needs to satisfy:

**Property 1** (Optimism). *With high probability, $Q_t(s,a) \leq Q^\star(s,a)$ holds for all $(s,a)$ and $t \geq 1$.*

**Property 2** (Recursion). *There exists a "bonus overhead" $\xi_H > 0$ and an absolute constant $d > 0$ such that the following holds with high probability:*

$$\sum_{t=1}^{T}(\mathring{Q}(s_t,a_t) - Q_t(s_t,a_t))_+ \leq \xi_H + \left(1 + \frac{d}{H}\right)\sum_{t=1}^{T}(\mathring{V}(s_t) - Q_t(s_t,a_t))_+,$$

*for $\mathring{Q} = Q_h^\star$ and $\mathring{V} = V_{h-1}^\star$ ($h = 1, \ldots, H$) as well as $\mathring{Q} = Q^\star$ and $\mathring{V} = V^\star$.[3]*

Property 1 is standard and can usually be ensured by using a certain "bonus" term derived from concentration equalities in the update. These bonus terms on $(s_t, a_t)$ accumulate into some bonus overhead in the final regret bound, which is exactly the role of $\xi_H$ in Property 2. In both of our algorithms, $\xi_H$ has a leading-order term $\tilde{\mathcal{O}}(\sqrt{B_\star SAC_K})$ and a lower-order term that increases in $H$.

Property 2 is a key property that provides a recursive form of the estimation error and allows us to connect it to the finite-horizon approximation. This is illustrated through the following two lemmas.

**Lemma 2.** *Property 2 implies $\sum_{t=1}^{T}(Q_H^\star(s_t,a_t) - Q_t(s_t,a_t))_+ \leq \mathcal{O}(H\xi_H)$.*

*Proof.* With $\mathring{Q} = Q_H^\star$ and $\mathring{V} = V_{H-1}^\star$, Property 2 implies

$$\sum_{t=1}^{T}(Q_H^\star(s_t,a_t) - Q_t(s_t,a_t))_+ \leq \xi_H + \left(1 + \frac{d}{H}\right)\sum_{t=1}^{T}(V_{H-1}^\star(s_t) - Q_t(s_t,a_t))_+$$

$$\leq \xi_H + \left(1 + \frac{d}{H}\right)\sum_{t=1}^{T}(Q_{H-1}^\star(s_t,a_t) - Q_t(s_t,a_t))_+,$$

where in the last step we use the optimality of $V_{H-1}^\star$ from Eq. (1). Repeatedly applying this argument, we eventually arrive at $\sum_{t=1}^{T}(Q_H^\star(s_t,a_t) - Q_t(s_t,a_t))_+ \leq H\left(1 + \frac{d}{H}\right)^H \xi_H + \left(1 + \frac{d}{H}\right)^H \sum_{t=1}^{T}(Q_0^\star(s_t,a_t) - Q_t(s_t,a_t))_+ = \mathcal{O}(H\xi_H)$, where the last step uses the facts $Q_0^\star(s_t,a_t) = 0$ and $\left(1 + \frac{d}{H}\right)^H \leq e^d$ (an absolute constant). $\square$

**Lemma 3.** *For any $\beta \in (0,1)$, if $H \geq \frac{4B_\star}{c_{\min}} \ln(2/\beta) + 1$, then Property 1 and Property 2 together imply $\sum_{t=1}^{T} Q^\star(s_t,a_t) - V^\star(s_t) = \mathcal{O}(\beta C_K + \xi_H)$.*

*Proof.* Applying Property 2 with $\mathring{Q} = Q^\star$ and $\mathring{V} = V^\star$, we have $\sum_{t=1}^{T}(Q^\star(s_t,a_t) - Q_t(s_t,a_t))_+ \leq \xi_H + \left(1 + \frac{d}{H}\right)\sum_{t=1}^{T}(V^\star(s_t) - Q_t(s_t,a_t))_+$. Now note that by Property 1, the Bellman optimality equation $V^\star(s_t) = \min_a Q^\star(s_t,a)$, and the fact $Q_t(s_t,a_t) = \min_a Q_t(s_t,a)$ (by the definition of $a_t$), the arguments within the clipping operation $(\cdot)_+$ are all non-negative and thus the clipping can be removed. Rearranging terms then gives

$$\sum_{t=1}^{T} Q^\star(s_t,a_t) - V^\star(s_t) \leq \xi_H + \frac{d}{H}\sum_{t=1}^{T}(V^\star(s_t) - Q_t(s_t,a_t))$$

$$\leq \xi_H + \frac{d}{H}\sum_{t=1}^{T}(Q^\star(s_t,a_t) - Q_t(s_t,a_t)). \qquad \text{(optimality of } V^\star\text{)}$$

---

[2] Note that our notation is perhaps unconventional compared to most works on finite-horizon MDPs, where $Q_h^\star$ usually refers to our $Q_{H-h}^\star$. We make this switch since we want to highlight the dependence on $H$ for $Q_H^\star$.

[3] Note that $\xi_H$ might be a random variable. In fact, it often depends on $C_K$.

It remains to bound the last term using the finite-horizon approximation $Q_H^\star$ as a proxy:

$$\sum_{t=1}^{T}(Q^\star(s_t, a_t) - Q_t(s_t, a_t)) = \sum_{t=1}^{T}(Q^\star(s_t, a_t) - Q_H^\star(s_t, a_t) + Q_H^\star(s_t, a_t) - Q_t(s_t, a_t))$$
$$= \mathcal{O}\left(TB_\star\beta + H\xi_H\right),$$

where the last step uses Lemma 1 and Lemma 2. Importantly, this term is finally scaled by $d/H$, which, together with the fact $\frac{TB_\star}{H} \le c_{\min}T \le C_K$, proves the claimed bound. $\quad\square$

Readers familiar with the literature might already recognize the term $\sum_{t=1}^{T} Q^\star(s_t, a_t) - V^\star(s_t)$ considered in Lemma 3, which is closely related to the regret. Indeed, with this lemma, we can conclude a regret bound for our generic algorithm.

**Theorem 1.** *For any $\beta \in (0, 1)$, if $H \ge \frac{4B_\star}{c_{\min}}\ln(2/\beta) + 1$, then Algorithm 1 ensures (with high probability) $R_K = \tilde{\mathcal{O}}\left(\sqrt{B_\star C_K} + B_\star + \beta C_K + \xi_H\right)$.*

*Proof.* We first decompose the regret as follows, which holds generally for any algorithm:

$$R_K = \sum_{k=1}^{K}\left(\sum_{i=1}^{I_k} c_i^k - V^\star(s_1^k)\right)$$
$$\le \sum_{k=1}^{K}\sum_{i=1}^{I_k}\left(c_i^k - V^\star(s_i^k) + V^\star(s_{i+1}^k)\right) = \sum_{t=1}^{T}(c_t - V^\star(s_t) + V^\star(s_t'))$$
$$= \sum_{t=1}^{T}(c_t - c(s_t, a_t)) + \sum_{t=1}^{T}(V^\star(s_t') - P_{s_t,a_t}V^\star) + \sum_{t=1}^{T}(Q^\star(s_t, a_t) - V^\star(s_t)). \quad (2)$$

The first and the second term are the sum of a martingale difference sequence (since $s_t'$ is drawn from $P_{s_t,a_t}$) and can be bounded by $\tilde{\mathcal{O}}\left(\sqrt{C_K}\right)$ and $\tilde{\mathcal{O}}\left(\sqrt{B_\star C_K} + B_\star\right)$ respectively using concentration inequalities; see Lemma 4, Lemma 35, and Lemma 5. The third term can be bounded using Lemma 3 directly, which finishes the proof. $\quad\square$

To get a sense of the regret bound in Theorem 1, first note that since $1/\beta$ only appears in a logarithmic term of the required lower bound of $H$, one can pick $\beta$ to be small enough so that the term $\beta C_K$ is dominated by others. Moreover, if $\xi_H$ is $\tilde{\mathcal{O}}(\sqrt{B_\star SAC_K})$ plus some lower-order term $\rho_H$ (which as mentioned is the case for our algorithms), then by solving a quadratic of $\sqrt{C_K}$, the regret bound of Theorem 1 implies $R_K = \tilde{\mathcal{O}}(B_\star\sqrt{SAK} + \rho_H)$, which is minimax optimal (ignoring $\rho_H$)!

Based on this analytical technique, it remains to design algorithms satisfying the two required properties. In the following sections, we provide two such examples, leading to the first model-free SSP algorithm and an improved model-based SSP algorithm.

## 4    The First Model-free Algorithm: LCB-Advantage-SSP

In this section, we present a model-free algorithm (the first in the literature) called LCB-Advantage-SSP that falls into our generic template and satisfies the required properties. It is largely inspired by the state-of-the-art model-free algorithm UCB-Advantage [Zhang et al., 2020b] for the finite-horizon problem. The pseudocode is shown in Algorithm 2, with only the lines instantiating the update rule of the $Q$ estimates numbered. Importantly, the space complexity of this algorithm is only $\mathcal{O}(SA)$ since we do not estimate the transition directly or conduct explicit finite-horizon reduction, and the time complexity is only $\mathcal{O}(1)$ in each step.

Specifically, for each state-action pair $(s, a)$, we divide the samples received when visiting $(s, a)$ into consecutive stages of exponentially increasing length, and only update $Q(s, a)$ at the end of a stage. The number of samples $e_j$ in stage $j$ is defined through $e_1 = H$ and $e_{j+1} = \lfloor(1 + 1/H)e_j\rfloor$ for some parameter $H$. Further define $\mathcal{L}^\star = \{E_j\}_{j\in\mathbb{N}^+}$ with $E_j = \sum_{i=1}^{j} e_i$, which contains all the indices indicating the end of some stage. As mentioned, the algorithm only updates $Q(s, a)$ when the

---

**Algorithm 2** LCB-ADVANTAGE-SSP

---

**Parameters:** horizon $H$, threshold $\theta^\star$, and failure probability $\delta \in (0,1)$.

**Define:** $\mathcal{L}^\star = \{E_j\}_{j \in \mathbb{N}^+}$ where $E_j = \sum_{i=1}^j e_i$, $e_1 = H$ and $e_{j+1} = \lfloor (1 + 1/H)e_j \rfloor$.

**Initialize:** $t \leftarrow 0$, $s_1 \leftarrow s_{\text{init}}$, $B \leftarrow 1$, for all $(s,a)$, $N(s,a) \leftarrow 0$, $M(s,a) \leftarrow 0$.

**Initialize:** for all $(s,a)$, $Q(s,a) \leftarrow 0$, $V(s) \leftarrow 0$, $V^{\text{ref}}(s) \leftarrow V(s)$, $\widehat{C}(s,a) \leftarrow 0$.

**Initialize:** for all $(s,a)$, $\mu^{\text{ref}}(s,a) \leftarrow 0$, $\sigma^{\text{ref}}(s,a) \leftarrow 0$, $\mu(s,a) \leftarrow 0$, $\sigma(s,a) \leftarrow 0$, $v(s,a) \leftarrow 0$.

**for** $k = 1, \dots, K$ **do**

  **repeat**

    Increment time step $t \overset{+}{\leftarrow} 1$.

    Take action $a_t = \operatorname{argmin}_a Q(s_t, a)$, suffer cost $c_t$, transit to and observe $s_t'$.

**1**    Increment visitation counters: $n = N(s_t, a_t) \overset{+}{\leftarrow} 1$, $m = M(s_t, a_t) \overset{+}{\leftarrow} 1$.

**2**    Update global accumulators: $\mu^{\text{ref}}(s_t, a_t) \overset{+}{\leftarrow} V^{\text{ref}}(s_t')$, $\sigma^{\text{ref}}(s_t, a_t) \overset{+}{\leftarrow} V^{\text{ref}}(s_t')^2$, $\widehat{C}(s_t, a_t) \overset{+}{\leftarrow} c_t$.

**3**    Update local accumulators: $v(s_t, a_t) \overset{+}{\leftarrow} V(s_t')$, $\mu(s_t, a_t) \overset{+}{\leftarrow} V(s_t') - V^{\text{ref}}(s_t')$, $\sigma(s_t, a_t) \overset{+}{\leftarrow} (V(s_t') - V^{\text{ref}}(s_t'))^2$.

**4**    **if** $n \in \mathcal{L}^\star$ **then**

**5**      Compute $\iota \leftarrow 256 \ln^6(4SAB_\star^8 n^5 / \delta)$, cost estimator $\widehat{c} = \frac{\widehat{C}(s_t, a_t)}{n}$, bonuses $b' \leftarrow 2\sqrt{\frac{B^2 \iota}{m}} + \sqrt{\frac{\widehat{c}\iota}{n}} + \frac{\iota}{n}$ and $b \leftarrow$

$$\sqrt{\frac{\sigma^{\text{ref}}(s_t, a_t)/n - (\mu^{\text{ref}}(s_t, a_t)/n)^2}{n}}\iota + \sqrt{\frac{\sigma(s_t, a_t)/m - (\mu(s_t, a_t)/m)^2}{m}}\iota + \left(\frac{4B}{n} + \frac{3B}{m}\right)\iota + \sqrt{\frac{\widehat{c}\iota}{n}}.$$

**6**      $Q(s_t, a_t) \leftarrow \max\left\{\widehat{c} + \frac{v(s_t, a_t)}{m} - b', Q(s_t, a_t)\right\}$.

**7**      $Q(s_t, a_t) \leftarrow \max\left\{\widehat{c} + \frac{\mu^{\text{ref}}(s_t, a_t)}{n} + \frac{\mu(s_t, a_t)}{m} - b, Q(s_t, a_t)\right\}$.

**8**      $V(s_t) \leftarrow \min_a Q(s_t, a)$.

**9**      **if** $V(s_t) > B$ **then** $B \leftarrow 2V(s_t)$.

**10**      Reset local accumulators: $v(s_t, a_t) \leftarrow 0$, $\mu(s_t, a_t) \leftarrow 0$, $\sigma(s_t, a_t) \leftarrow 0$, $M(s_t, a_t) \leftarrow 0$.

**11**    **if** $\sum_a N(s_t, a)$ *is a power of two not larger than* $\theta^\star$ **then** $V^{\text{ref}}(s_t) \leftarrow V(s_t)$.

    **if** $s_t' \neq g$ **then** $s_{t+1} \leftarrow s_t'$; **else** $s_{t+1} \leftarrow s_{\text{init}}$, **break**.

---

total number of visits to $(s,a)$ falls into the set $\mathcal{L}^\star$ (Line 4). The algorithm also maintains an estimate $V$ for $V^\star$, which always satisfies $V(s) = \min_a Q(s,a)$ (Line 8), and importantly another reference value function $V^{\text{ref}}$ whose role and update rule are to be discussed later.

In addition, some local and global accumulators are maintained in the algorithm. Local accumulators only store information related to the current stage. These include: $M(s,a)$, the number of visits to $(s,a)$ within the current stage; $v(s,a)$, the cumulative value of $V(s')$ within the current stage, where $s'$ represents the next state after each visit to $(s,a)$; and finally $\mu(s,a)$ and $\sigma(s,a)$, the cumulative values of $V(s') - V^{\text{ref}}(s')$ and its square respectively within the current stage (Line 3). These local accumulators are reset to zero at the end of each stage (Line 10).

On the other hand, global accumulators store information related to all stages and are never reset. These include: $N(s,a)$, the number of visits to $(s,a)$ from the beginning; $\widehat{C}(s,a)$, total cost incurs at $(s,a)$ from the beginning; and $\mu^{\text{ref}}(s,a)$ and $\sigma^{\text{ref}}(s,a)$, the cumulative value of $V^{\text{ref}}(s')$ and its square respectively from the beginning, where again $s'$ represents the next state after each visit to $(s,a)$ (Line 2).

We are now ready to describe the update rule of $Q$. The first update, Line 6, is intuitively based on the equality $Q^\star(s,a) = c(s,a) + P_{s,a}V^\star$ and uses $v(s,a)/M(s,a)$ as an estimate for $P_{s,a}V^\star$ together with a (negative) bonus $b'$ derived from Azuma's inequality (Line 5). As mentioned, the bonus is necessary to ensure Property 1 (optimism) so that $Q$ is always a lower confidence bound of $Q^\star$ (hence the name "LCB"). Note that this update only uses data from the current stage (roughly $1/H$ fraction of the entire data collected so far), which leads to an extra $\sqrt{H}$ factor in the regret.

To address this issue, Zhang et al. [2020b] introduce a variance reduction technique via a reference-advantage decomposition, which we borrow here leading to the second update rule in Line 7. This is intuitively based on the decomposition $P_{s,a}V^\star = P_{s,a}V^{\text{ref}} + P_{s,a}(V^\star - V^{\text{ref}})$, where $P_{s,a}V^{\text{ref}}$ is approximated by $\mu^{\text{ref}}(s,a)/N(s,a)$ and $P_{s,a}(V^\star - V^{\text{ref}})$ is approximated by $\mu(s,a)/M(s,a)$. In addition, a "variance-aware" bonus term $b$ is applied, which is derived from a tighter Freedman's inequality (Line 5). The reference function $V^{\text{ref}}$ is some snapshot of the past value of $V$, and is guaranteed to be $\mathcal{O}(c_{\min})$ close to $V^\star$ on a particular state as long as the number of visits to this state exceeds some threshold $\theta^\star = \tilde{\mathcal{O}}\left(B_\star^2 H^3 SA/c_{\min}^2\right)$ (Line 11). Overall, this second update rule not only removes the extra $\sqrt{H}$ factor as in [Zhang et al., 2020b], but also turns some terms of order $\tilde{\mathcal{O}}(\sqrt{T})$ into $\tilde{\mathcal{O}}(\sqrt{C_K})$ in our context, which is important for obtaining the optimal regret.

Despite the similarity, we emphasize several key differences between our algorithm and that of [Zhang et al., 2020b]. First, [Zhang et al., 2020b] maintains a different $Q$ estimate for each step of an episode (which is natural for a finite-horizon problem), while we only maintain one $Q$ estimate (which is natural for SSP). Second, we update the reference function $V^{\text{ref}}(s)$ whenever the number of visits to $s$ doubles (while still below the threshold $\theta^\star$; see Line 11), instead of only updating it once as in [Zhang et al., 2020b]. We show in Lemma 8 that this helps reduce the sample complexity and leads to a smaller lower-order term in the regret. Third, since there is no apriori known upper bound on $V$ (unlike the finite-horizon setting), we maintain an empirical upper bound $B$ (in a doubling manner) such that $V(s) \leq B \leq 2B_\star$ (Line 9), which is further used in computing the bonus terms $b$ and $b'$. This is important for eventually developing a parameter-free algorithm.

In Appendix D, we show that Algorithm 2 indeed satisfies the two required properties.

**Theorem 2.** *Let* $H = \left\lceil \frac{4B_\star}{c_{\min}} \ln(\frac{2}{\beta}) + 1 \right\rceil_2$ *for* $\beta = \frac{c_{\min}}{2B_\star^2 SAK}$ *and* $\theta^\star = \tilde{\mathcal{O}}\left(\frac{B_\star^2 H^3 SA}{c_{\min}^2}\right)$ *be defined in Lemma 8, then Algorithm 2 satisfies Property 1 and Property 2 with* $d = 3$ *and* $\xi_H = \tilde{\mathcal{O}}\left(\sqrt{B_\star SAC_K} + \frac{B_\star^2 H^3 S^2 A}{c_{\min}}\right)$.

*Proof Sketch.* The proof of Property 1 largely follows the analysis of [Zhang et al., 2020b, Proposition 4] for the designed bonuses. To prove Property 2, similarly to [Zhang et al., 2020b] we can show:

$$\sum_{t=1}^{T} (\mathring{Q}(s_t, a_t) - Q_t(s_t, a_t))_+ \lesssim \xi_H + \sum_{t=1}^{T} \frac{1}{m_t} \sum_{i=1}^{m_t} P_{s_{\check{l}_{t,i}}, a_{\check{l}_{t,i}}}(\mathring{V} - V_{\check{l}_{t,i}})_+,$$

where $m_t$ is the value of $m$ used in computing $Q_t(s_t, a_t)$, and $\check{l}_{t,i}$ is the $i$-th time step the agent visits $(s_t, a_t)$ among those $m_t$ steps. Now it suffices to show that $\sum_{t=1}^{T} \frac{1}{m_t} \sum_{i=1}^{m_t} P_{s_{\check{l}_{t,i}}, a_{\check{l}_{t,i}}}(\mathring{V} - V_{\check{l}_{t,i}})_+ \lesssim (1 + \frac{3}{H}) \sum_{t=1}^{T} (\mathring{V}(s_t) - V_t(s_t))_+$, which is proven in Lemma 13. $\square$

As a direct corollary of Theorem 1, we arrive at the following regret guarantee.

**Theorem 3.** *With the same parameters as in Theorem 2, with probability at least* $1 - 60\delta$, *Algorithm 2 ensures* $R_K = \tilde{\mathcal{O}}\left(B_\star\sqrt{SAK} + \frac{B_\star^5 S^2 A}{c_{\min}^4}\right)$.

We make several remarks on our results. First, while Algorithm 2 requires setting the two parameters $H$ and $\theta^\star$ in terms of $B_\star$ to obtain the claimed regret bound, one can in fact achieve the exact same bound without knowing $B_\star$ by slightly changing the algorithm. The high level idea is to first apply the doubling trick from Tarbouriech et al. [2021b] to determine an upper bound on $B_\star$, then try logarithmically many different values of $H$ and $\theta^\star$ simultaneously, each leading to a different update rule for $Q$ and $V^{\text{ref}}$. This only increases the time and space complexity by a logarithmic factor, without hurting the regret (up to log factors). Details are deferred to Section D.5.

Second, as mentioned in Section 2, when $c_{\min}$ is unknown or $c_{\min} = 0$, one can clip all observed costs to $\epsilon$ if they are below $\epsilon > 0$, which introduces an additive regret term of order $\mathcal{O}(\epsilon K)$. By picking $\epsilon$ to be of order $K^{-1/5}$, our bound becomes $\tilde{\mathcal{O}}(K^{4/5})$ ignoring other parameters. Although most existing works suffer the same issue, this is certainly undesirable, and our second algorithm to be introduced in the next section completely avoids this issue by having only logarithmic dependence on $1/c_{\min}$.

---
**Algorithm 3** SVI-SSP
---
**Parameters:** horizon $H$, value function upper bound $B$, and failure probability $\delta \in (0, 1)$.

**Define:** $\mathcal{L} = \{E_j\}_{j \in \mathbb{N}^+}$, where $E_j = \sum_{i=1}^{j} e_i, e_j = \lfloor \widetilde{e}_j \rfloor$, and $\widetilde{e}_1 = 1, \widetilde{e}_{j+1} = \widetilde{e}_j + \frac{1}{H} e_j$.

**Initialize:** $t \leftarrow 0, s_1 \leftarrow s_{\text{init}}$.

**Initialize:** for all $(s, a, s'), n(s, a, s') \leftarrow 0, n(s, a) \leftarrow 0, Q(s, a) \leftarrow 0, V(s) \leftarrow 0, \widehat{C}(s, a) \leftarrow 0$.

**for** $k = 1, \ldots, K$ **do**

    **repeat**

        Increment time step $t \overset{+}{\leftarrow} 1$.

        Take action $a_t = \operatorname{argmin}_a Q(s_t, a)$, suffer cost $c_t$, transit to and observe $s'_t$.

1        Update accumulators: $n = n(s_t, a_t) \overset{+}{\leftarrow} 1, n(s_t, a_t, s'_t) \overset{+}{\leftarrow} 1, \widehat{C}(s_t, a_t) \overset{+}{\leftarrow} c_t$.

2        **if** $n \in \mathcal{L}$ **then**

3            Update empirical transition: $\bar{P}_{s_t, a_t}(s') \leftarrow \frac{n(s_t, a_t, s')}{n}$ for all $s'$.

4            Compute $\iota \leftarrow 20 \ln \frac{2SAn}{\delta}$, cost estimator $\widehat{c} \leftarrow \frac{\widehat{C}(s,a)}{n}$, and bonus $b \leftarrow \max\left\{ 7\sqrt{\frac{\mathbb{V}(\bar{P}_{s_t, a_t}, V)\iota}{n}}, \frac{49B\iota}{n} \right\} + \sqrt{\frac{\widehat{c}\iota}{n}}$.

5            $Q(s_t, a_t) \leftarrow \max\{\widehat{c} + \bar{P}_{s_t, a_t} V - b, Q(s_t, a_t)\}$.

6            $V(s_t) \leftarrow \operatorname{argmin}_a Q(s_t, a)$.

        **if** $s'_t \neq g$ **then** $s_{t+1} \leftarrow s'_t$; **else** $s_{t+1} \leftarrow s_{\text{init}}$, **break**.

---

Finally, we point out that, just as in the finite-horizon case, the variance reduction technique is crucial for obtaining the minimax optimal regret. For example, if one instead uses an update rule similar to the (suboptimal) Q-learning algorithm of [Jin et al., 2018], then this is essentially equivalent to removing the second update (Line 7) of our algorithm. While this still satisfies Property 2, the bonus overhead $\xi_H$ would be $\sqrt{H}$ times larger, resulting in a suboptimal leading term in the regret.

## 5 An Optimal and Efficient Model-based Algorithm: SVI-SSP

In this section, we propose a simple model-based algorithm called SVI-SSP (Sparse Value Iteration for SSP) following our template, which not only achieves the minimax optimal regret even when $c_{\min} = 0$, matching the state-of-the-art by a recent work [Tarbouriech et al., 2021b], but also admits highly sparse updates, making it more efficient than all existing model-based algorithms. The pseudocode is in Algorithm 3, again with only the lines instantiating the update rule for $Q$ numbered.

Similar to Algorithm 2, SVI-SSP divides samples of each $(s, a)$ into consecutive stages of (roughly) exponentially increasing length, and only update $Q(s, a)$ at the end of a stage (Line 2). However, the number of samples $e_j$ in stage $j$ is defined slightly differently through $e_j = \lfloor \widetilde{e}_j \rfloor, \widetilde{e}_1 = 1$, and $\widetilde{e}_{j+1} = \widetilde{e}_j + \frac{1}{H} e_j$ for some parameter $H$. In the long run, this is almost the same as the scheme used in Algorithm 2, but importantly, it forces more frequent updates at the beginning — for example, one can verify that $e_1 = \cdots = e_H = 1$, meaning that $Q(s, a)$ is updated every time $(s, a)$ is visited for the first $H$ visits. This slight difference turns out to be important to ensure that the lower-order term in the regret has no poly($H$) dependence, as shown in Lemma 16 and further discussed in Remark 3. More intuition on the design of this update scheme is provided in Section E.1.

The update rule for $Q$ is very simple (Line 5). It is again based on the equality $Q^\star(s, a) = c(s, a) + P_{s,a}V^\star$, but this time uses $\bar{P}_{s,a}V - b$ as an approximation for $P_{s,a}V^\star$, where $\bar{P}_{s,a}$ is the empirical transition directly calculated from two counters $n(s, a)$ and $n(s, a, s')$ (number of visits to $(s, a)$ and $(s, a, s')$ respectively), $V$ is such that $V(s) = \min_a Q(s, a)$, and $b$ is a special bonus term (Line 4) adopted from [Tarbouriech et al., 2021b, Zhang et al., 2020a] which ensures that $Q$ is an optimistic estimate of $Q^\star$ and also helps remove poly($H$) dependence in the regret.

SVI-SSP exhibits a unique structure compared to existing algorithms. In each update, it modifies only one entry of $Q$ (similarly to model-free algorithms), while other model-based algorithms such as [Tarbouriech et al., 2021b] perform value iteration for every entry of $Q$ repeatedly until convergence (concrete time complexity comparisons to follow). We emphasize that our implicit finite-horizon

analysis is indeed the key to enable us to derive a regret guarantee for such a sparse value iteration algorithm. Specifically, in Appendix E, we show that SVI-SSP satisfies the two required properties.

**Theorem 4.** *If $B \geq B_\star$ and $H = \lceil \frac{4B}{c_{\min}} \ln(\frac{2}{\beta}) + 1 \rceil_2$ for $\beta = \frac{c_{\min}}{2B^2 SAK}$, then Algorithm 3 satisfies Property 1 and Property 2 with $d = 1$ and $\xi_H = \tilde{\mathcal{O}}(\sqrt{B_\star SAC_K} + BS^2 A + \beta C_K)$, where the dependence on $H$ in $\xi_H$ is hidden in logarithmic terms.*

*Proof Sketch.* The proof of Property 1 largely follows the analysis of [Tarbouriech et al., 2021b, Lemma 15]. To prove Property 2, we first show $\sum_{t=1}^{T} (\mathring{Q}(s_t, a_t) - Q_t(s_t, a_t))_+ \lesssim \xi_H + \sum_{t=1}^{T} P_t(\mathring{V} - V_{l_t})_+$, where $l_t$ is the last time step $Q(s_t, a_t)$ is updated. Then, the remaining main steps are shown below with all details deferred to the corresponding key lemmas:

$$\sum_{t=1}^{T} P_t(\mathring{V} - V_{l_t})_+ \lesssim \left(1 + \frac{1}{H}\right) \sum_{t=1}^{T} P_t(\mathring{V} - V_t)_+ \qquad \text{(Lemma 16)}$$

$$\lesssim \left(1 + \frac{1}{H}\right) \sum_{t=1}^{T} (\mathring{V}(s_t) - V_t(s_t))_+ + \left(1 + \frac{1}{H}\right) \sum_{t=1}^{T} (P_t - \mathbb{I}_{s'_t})(\mathring{V} - V_t)_+$$

$$\lesssim \left(1 + \frac{1}{H}\right) \sum_{t=1}^{T} (\mathring{V}(s_t) - V_t(s_t))_+ + \xi_H, \qquad \text{(Lemma 22 and Lemma 21)}$$

which completes the proof. $\qquad \square$

Again, as a direct corollary of Theorem 1, we arrive at the following regret guarantee.

**Theorem 5.** *With the same parameters as in Theorem 4, with probability at least $1 - 12\delta$, Algorithm 3 ensures $R_K = \tilde{\mathcal{O}}(B_\star \sqrt{SAK} + BS^2 A)$.*

Setting $B = B_\star$, our bound becomes $\tilde{\mathcal{O}}(B_\star \sqrt{SAK} + B_\star S^2 A)$, which is minimax optimal even when $c_{\min}$ is unknown or $c_{\min} = 0$ (this is because the dependence on $1/c_{\min}$ is only logarithmic, and one can clip all observed costs to $\epsilon$ if they are below $\epsilon = 1/K$ in this case without introducing poly($K$) overhead to the regret). When $B_\star$ is unknown, we can use the same doubling trick from Tarbouriech et al. [2021b] to obtain almost the same bound (with only the lower-order term increased to $\tilde{\mathcal{O}}(B_\star^3 S^3 A)$); see Section E.5 for details.[4]

**Comparison with EB-SSP [Tarbouriech et al., 2021b]** Our regret bounds match exactly the state-of-the-art by Tarbouriech et al. [2021b]. Thanks to the sparse update, however, SVI-SSP has a much better time complexity. Specifically, for SVI-SSP, each $(s, a)$ is updated at most $\tilde{\mathcal{O}}(H) = \tilde{\mathcal{O}}(B_\star/c_{\min})$ times (Lemma 16), and each update takes $\mathcal{O}(S)$ time, leading to total complexity $\tilde{\mathcal{O}}(B_\star S^2 A/c_{\min})$. On the other hand, for EB-SSP, although each $(s, a)$ only causes $\tilde{\mathcal{O}}(1)$ updates, each update runs value iteration on all entries of $Q$ until convergence, which takes $\tilde{\mathcal{O}}(B_\star^2 S^2/c_{\min}^2)$ iterations (see their Appendix C) and leads to total complexity $\tilde{\mathcal{O}}(B_\star^2 S^5 A/c_{\min}^2)$, much larger than ours.

**Comparison with ULCVI [Cohen et al., 2021]** Another recent work by Cohen et al. [2021] using explicit finite-horizon approximation also achieves minimax regret but requires the knowledge of some hitting time of the optimal policy. Without this knowledge, their bound has a large $1/c_{\min}^4$ dependence in the lower-order term just as our model-free algorithm. Our results in this section show that implicit finite-horizon approximation has advantage over explicit approximation apart from reducing space complexity: the former does not necessarily introduce poly($H$) dependence even for the lower-order term, while the latter does under the current analysis.

## Acknowledgments and Disclosure of Funding

LC thanks Chen-Yu Wei for many helpful discussions. HL is supported by NSF Award IIS-1943607 and a Google Faculty Research Award. MJ and RJ's research is supported by NSF CCF-1817212, NSF ECCS-1810447 and ONR N00014-20-1-2258 awards.

---

[4]We note that this doubling trick is in fact also applicable to Algorithm 2. However, the specific approach we propose for this algorithm in Section D.5 is better in the sense that it does not worsen the regret at all.

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
