# OpenReview forum: "Implicit Finite-Horizon Approximation and Efficient Optimal Algorithms for Stochastic Shortest Path"
_NeurIPS.cc/2021/Conference — NeurIPS 2021 Poster_

### Official Review · Reviewer_qWsp · 2021-06-25

**Rating:** 7
**Confidence:** 4

**Summary:**

The authors propose a generic template for regret minimization algorithms in SSP. They develop the first efficient model-free SSP algorithm. They introduced a technique called implicit finite-horizon approximation, which approximates the SSP model by a finite-horizon counterpart implicitly.

**Limitations And Societal Impact:**

For SSP, one specifically important question is why the algorithm halts (i.e. reach the goal state) in finite time with high probability for each episode. I did not see any discussion on this point in the paper. Even though this follows from the finite regret of the algorithm, the authors should elaborate this point explicitly in the main text.


**Main Review:**

Originality: This paper proposed a technique that can be used to analyze the SSP problem. Although the major contribution, which to me is the algorithm LCB-advantage, borrows idea from the well known result of Zhang et al 2020, adpating this result to the SSP setting is still a non-trivial task in my opinion. Also, this should be the first model-free algorithm for SSP. So I think this is a successful adaption of the known technique to a new problem.

Quality: The technical part of the paper looks correct.

Clarity: The paper is very clearly written and the idea is well presented.

Significance: This is a good contribution to the SSP problem which is an important direction in theoretical reinforcement learning. The proposed algorithm as I know is the first model-free algorithm for SSP.

**Time Spent Reviewing:**

3

---

> ### Author Response · Authors · 2021-08-09
> **Response to reviewer qWsp**
>
> Thanks for your valuable feedback.
>
>
>
> - **Why the algorithm halts (i.e. reach the goal state) in finite time with high probability**: intuitively, as the Q estimate gets more and more accurate, our policy behaves more and more closely to the optimal policy, which then guarantees reaching the goal with high probability. On the other hand, at the beginning of the learning process, the algorithm indeed can take a long time to reach the goal, which is also what we observed in experiments. In the analysis, the number of total steps is bounded as $T\leq \frac{C_K}{c_{\min}}$ and we then show the total cost $C_K$ is bounded (recall that when $c_{\min}$ is zero, costs are perturbed such that the effective $c_{\min}$ is non-zero). We will add more discussions on this in the revision.

---

### Official Review · Reviewer_HJ7d · 2021-07-13

**Rating:** 7
**Confidence:** 4

**Summary:**

This paper presents an algorithmic template for the SSP model that unifies algorithms based on value optimism. Then, two instances of the template are presented and analyzed. The first instance is a model-free algorithm (the first for SSP) that achieves optimal regret but only when all costs are strictly positive. The second is a model-based algorithm that achieves optimal regret without any assumption on the cost. This algorithm enjoys improved computational complexity by performing 1-step planning instead of full-planning (also the first for SSP).

**Limitations And Societal Impact:**

yes

**Main Review:**

Overall this is a really good paper but I need to mention that I did not have the time to fully check the proofs in the appendix.
The theoretical guarantees of the algorithms are strong, and their computational and space complexity are especially impressive (I have some concerns regarding computational complexity, see next).
The technical contribution is also promising. The implicit usage of finite-horizon is able to benefit both from recent reductions to finite-horizon and from parameter-free and horizon-free guarantees of Tarbouriech et al. [2021b].

However, there are still quite a few things that I think could be presented better/improved in the final version of this paper. I think that they can be fixed after the rebuttal, but I stress that these should be addressed if the authors wish for this paper to really have an impact.

While the paper is beautifully written up until section 4, the rest of it is hard to follow and does not focus on the right things in my opinion. For starters, while algorithm 1 is very useful, I found algorithms 2,3 highly confusing since they are too technical. I would remove them and keep just the explanations that you have in the text. After all, they just specify the exploration bonuses and related computations that are familiar to most readers from other RL settings. For this reason, I would not have focused on the description of these computations, I think that they are much less interesting in the context of SSP. Instead, I think the focus should be on how to prove properties 1 and 2 for these two algorithms, i.e., how do you take these finite-horizon techniques and use them in the context of SSP. This is not described at all, and I think it is the most important contribution after you already explained so nicely the template of the algorithm and analysis. On a related note, maybe there is a better way (could be less formal) to define $e_j$, $\tilde e_j$ and $E_j$? it is hard to comprehend these definitions and they don't give the intuition about this exponentially growing stage lengths.

I have some concerns regarding $c_{min} $. First, the computational complexity grows with $1/ c_{min} $. when $c_{min} = 0$ this becomes polynomial in $K$. I know that this is similar to previous work, but here you state that your algorithms are very efficient and admit highly sparse updates. This is certainly not the case when $c_{min} = 1/poly(K) $ so you should be more accurate about that. Second, again when $c_{min} $ is small the regret of the model-free algorithm has a huge additive term that will be dominant in many cases. My main question here is (this one also goes for the complexity): Can you fix $H $ to be smaller than $B_* / c_{min} $ and get alternative guarantees? With explicit reduction to finite-horizon this is possible, and I think this is crucial for your result to be more meaningful in the general case. A related concern here is if this template is completely "implicit". This is definitely a big step towards implicit analysis of SSP using finite-horizon, but since $H $ is an explicit parameter, there is more work that could be done here and I think that this is something that should be discussed in order to encourage future work on the subject.

Additional concerns:
1.  I think that the empirical evaluation only hurts the results of this paper. Since experiments are not standard in these regret minimization papers, their results should be more clear. I think the experiments presented here only draw theory and practice further apart in the context of RL, as they show that the "simple" algorithms are good enough. However, this is due to picking the "wrong" examples. Also, note that in grid world $c_{min} = 1 $! For random MDP what was $c_{min} $?

2. I have a small concern regarding the implicit finite-horizon analysis. $\xi_H $ is a random variable that depends on the total cost of the agent in $K $ episodes. However, the analysis refers to it as something that depends solely on $H $. I did not find a problem in the analysis, but I am a little concerned about this (as $\xi_H $ is also not bounded). I think that denoting it as a function of the total cost could make this part more clear and avoid any issues. Also, $\delta $ needs to be picked small enough (also as a function of the agent's total cost) and this is not mentioned in the theorems.

3. Recent SSP papers did not assume deterministic cost function. I think this paper should address stochastic cost function explicitly to match them.

Post-rebuttal: I am keeping my positive score. I hope that the important clarifications discussed with Reviewer WgWg will be incorporated in the camera-ready version.

**Time Spent Reviewing:**

3.5

---

> ### Author Response · Authors · 2021-08-09
> **Response to reviewer HJ7d**
>
> Thanks for your valuable feedback. Please find below detailed response to your comments:
>
>
>
> - **The focus should be on how to prove properties 1 and 2 for these two algorithms:** We agree that there is not much on how to prove these properties in the main text of the current version. In the revision, we will discuss key ideas of these proofs: for Property 1 (optimism), the proof largely follows applications of corresponding bonuses in the finite horizon setting, while for Property 2 (recursion), the key step is obtaining the recursive form with the $1+\frac{1}{H}$ factor, which corresponds to Lemma 11 for the model-free algorithm and Lemma 14 for the model-based algorithm. We will include intuition on these lemmas and how they are applied to prove Property 2 in the revision.
>
>
> - **A better explanation on $e_j$, $\widetilde{e}_j$ and $E_j$:** In our model-based algorithm, $\widetilde{e}_j$ is the "potential" length of stage $j$, while $e_j=\lfloor \widetilde{e}_j \rfloor$ is the actual length of stage $j$.
>
> Based on the update rule $\widetilde{e}_{j+1}=\widetilde{e}_j+\frac{1}{H}e_j\approx (1+\frac{1}{H})\widetilde{e}_j$, both lengths grow exponentially with a factor of $1+\frac{1}{H}$.
>
> See also Appendix C.1 for more intuition on this update scheme. The main reason that we need a slightly modified update scheme in the model-based case is that we want $e_1=1$ instead of $H$ to avoid the $H$ dependency in the lower order term. Indeed, for the simpler update $e_{j+1} = \lfloor (1+\frac{1}{H}) e_j \rfloor$ of the model-free algorithm, $e_1$ needs to be at least $H$ to ensure that the stage length grows exponentially.
>
> - **Can we fix $H$ to be smaller than $B_{\star}/c_{\min}$ and get an alternative guarantee?** Unfortunately, our current analysis does not support $H$ to be a smaller quantity such as $T_{\star}=\max_sT^{\pi^{\star}}(s)$, which on the other hand is allowed by an explicit finite horizon reduction. In order to set $H$ in terms of $T_{\star}$, we need the expected cost of the optimal policy in the approximated MDP to be an upper bound of that in the original MDP. This is not true in our case, since we approximate an SSP instance by its truncated version. Enabling a smaller value of $H$ is definitely an interesting future direction.
>
> - **$H$ is an explicit parameter, can it be completely implicit?** We can make it implicit in some sense using the parameter-free version of our algorithms described in Sections B.5 and C.5.
>
> - **What's $c_{\min}$ in RandomMDP:** For our random environment, $c_{\min}$ is around $0.04$, and $B_{\star}$ is around $1.5$. Our experiment on this environment does seem to indicate the benefit of the more advanced algorithms that are minimax optimal despite $c_{\min}$ being tiny.
>
> - **Concern on $\xi_H$:** In the revision, we will emphasize that $\xi_H$ is a random variable and it depends on $C_K$.
>
> - **$\delta$ should also be set small enough, as a function of $C_K$**: We assume that you are referring to the $\delta$ used in Theorem 1. In the later algorithm sections, we use $\beta$ in place of $\delta$ when invoking Theorem 1, and we use a small enough $\beta$ to ensure that $\beta C_K$ is small (see its value in Theorems 2 and 4). Note that $\delta$ in Theorem 1 is not the same as the failure probability, which is also referred to as $\delta$ in the later algorithm sections and does not need to be set w.r.t. the total costs of the learner. In the revision, we will change $\delta$ to $\beta$ in Theorem 1 to avoid confusion.
>
> - **Stochastic costs instead of deterministic costs**: Indeed, this is a good suggestion. Please also see our reply "Generalization to unknown stochastic costs" to reviewer U8Kt.

---

### Official Review · Reviewer_WgWg · 2021-07-16

**Rating:** 7
**Confidence:** 4

**Summary:**

The paper introduces a generic template for regret minimization in SSP (stochastic shortest path). It instantiates it with 1) a model-free algorithm (the first for SSP) that achieves the minimax rate under positive costs, and 2) a model-based algorithm that matches the best existing regret rate and admits sparse updates.

**Limitations And Societal Impact:**

Yes

**Main Review:**

The paper is clear, well written and well executed. While other works have also recently studied regret minimization for tabular SSP, the submission makes some relevant contributions on this setting.

The paper proposes a generic template for regret minimization for SSP based on an implicit finite-horizon approximation. The template is sound, easy to follow and allows to decompose the analysis between algorithm-agnostic aspects and algorithm-specific aspects.
The paper mentions that explicit finite-horizon approximation (as done in prior works) “unavoidably introduces poly(H) dependence even for the lower-order term”: could the authors explain why this is the case? (i.e., is there a reason why using horizon-free techniques in the finite-horizon reduction would prevent obtaining a horizon-free bound for SSP?).

First, the template is instantiated to obtain the first model-free algorithm for SSP. This is a strong contribution. Indeed, the analysis of model-free SSP induces the technical challenge of how to handle the error propagation across time steps (this does not appear in the layered structure of finite-horizon MDPs). The proposed template can nicely handle this propagation.
While the algorithm is largely inspired from UCB-Advantage [Zhang et al., 2020b], it performs a few necessary modifications and can be made parameter-free. The authors mention that their “smoother” update of the $V^{\text{ref}}$ function leads to a lower-order term in the regret: out of curiosity, could you quantify the improvement obtained by this technique (both in SSP and in finite-horizon MDPs)?
The regret bound is minimax optimal when costs are positive. While there is a poor additive dependence on $1/c_{\min}$ which leads to a general $K^{4/5}$ bound, this issue of poor lower order terms w.r.t. the horizon appears in all existing model-free algorithms. Does the extension to unknown stochastic costs (mentioned in footnote 1) require the knowledge of $c_{\min}$ (which appears in the definition of the horizon $H$ in Algorithm 2) or can it be somehow adaptive to it?

Second, the template is instantiated in a model-based fashion. The proposed algorithm and analysis are tightly connected to that of the recent algorithm EB-SSP [Tarbouriech et al., 2021b] (e.g., similar exploration bonus, recursion-based analysis, parameter-free scheme), and the obtained regret bound is the same. The main difference lies in the focus on the time complexity (with one-step updates), which is an interesting component w.r.t. existing SSP algorithms (and most other model-based algorithms) that perform full-planning. For instance the update scheme is novel (compared to the one in [Zhang et al., 2020a]) and allows to maintain the horizon-free regret.

Minor comment/suggestion: there are some parts where the authors could perhaps be slightly more precise in the presentation for non-familiar readers. It could be mentioned on top of page 13 that the $\widetilde{O}$ notation in the analysis usually includes $\log(T)$ terms, with $T$ the total time steps which is a random variable in SSP but can be easily handled under positive costs. Also the passage from $c_{\min}$ > 0 to $c_{\min}$ = 0, while standard and already performed in prior works, could be more carefully written: indeed as explained in [Tarbouriech et al., 2021b], doing the cost perturbation leads to different final bounds (some horizon-free, others not) depending on the prior knowledge available: in this sense, the statement in introduction “SVI-SSP achieves minimax regret $\widetilde{O}(B \sqrt{SAK} + B_{\star} S^2 A)$ even when $c_{\min} = 0$” (line 44-45) could be clarified, as it seems this requires either knowledge of a loose estimate of $T_{\star}$ or that $K \gtrsim T_{\star}$. The paper could perhaps report in the appendix more complete bounds, arguments or references to specific passages of prior works that share them (so as not to crowd the main paper with quite standard technicalities already discussed in previous works).

**Time Spent Reviewing:**

4.5

---

> ### Author Response · Authors · 2021-08-09
> **Response to reviewer WgWg**
>
> Thanks for your valuable feedback. We will incorporate all minor comments on the writings. Please find below detailed response to your other comments:
>
>
>
> - **Why explicit finite horizon approximation introduces poly(H) dependency?** In prior work [Cohen et al., 2021], the $H$ dependency in the lower order term mainly comes from the approximation of SSP by a finite horizon MDP, instead of from the regret bound of the finite horizon algorithm (whose lower order term can be horizon-free). More specifically, in their analysis, they mark each state-action pair as "known" or "unknown". When an unknown state-action pair is visited, they treat it as a bad event and bound the total costs of the current episode by $H$, which unavoidably introduces the poly(H) dependency in the lower order term.
>
>
>
> - **Quantify the improvement obtained by "smoother" update of $V^{\text{ref}}$:** If we only update the reference value once, the cost of reaching precision $\epsilon$ is of order $O(\frac{1}{\epsilon^2})$. With our "smoother" update, we improve the sample complexity to $O(\frac{1}{\epsilon})$ (see also Lemma 6 for more details). This translates to the following improvement: for SSP, the lower order term is $B_{\star}^5S^2A/c_{\min}^4$ with our new update, and $B_{\star}^5S^2A/c_{\min}^5$ without; for finite-horizon MDPs, the lower order term is $H^{6.5}S^2A$ with our new update, and $H^{7}S^2A$ without (as in Zhang et al. [2020b]).
>
>
>
> - **Does the extension to unknown stochastic costs require the knowledge of $c_{\min}?$**: Please see our response "Generalization to unknown stochastic costs" to reviewer U8Kt. In a word, for the model-based algorithm, the answer is no, while for the model-free algorithm, the answer is yes in order to maintain the same bound.

---

> > ### Comment · Reviewer_WgWg · 2021-08-17
> > **Response to Rebuttal**
> >
> > Thanks for the answers. I have some additional comments and questions:
> > - **Minor follow-up comment on explicit finite-horizon approximation:** I agree that the analysis of explicit finite-horizon reduction in [Cohen et al., 2021] must give a poly(H) dependence due to the known/unknown state-action pairs distinction. I'm rather wondering whether this is just a question of analysis, or whether the fact itself of performing explicit finite-horizon reduction must unavoidably give poly(H) dependency (even with the tightest analysis imaginable). It is fine if the authors do not have a definite answer, but if this is the case I would simply encourage to rephrase lines 336-339 and make clearer the distinction between "current analysis of" and "fundamental idea of" when referring to the "explicit approximation".
> > - **Shared comment with Reviewer HJ7d on $c_{\min}$/efficiency**: I agree with Reviewer HJ7d's concern on $c_{\min}$ and computational complexity. In the general cost case, if $c_{\min}$ must be set inversely to $K$, this seems to imply $\Omega(K)$ complexity. If so, the authors should clarify and discuss this limitation in the main paper (especially because an important focus and claim of the paper is about efficiency).
> > - **New concern/question on the parameter-free algorithm for LCB-ADVANTAGE-SSP:** From what I understand, to tune the number of update rules in Appendix B.5, the algorithm must rely on the assumption (stated page 13) that "$1 \leq B_{\star} \leq \sqrt{K}$ (in general, a known constant $\alpha > 0$ such that $B_{\star} \leq K^{\alpha}$ suffices)." However, making this assumption seems to defeat the parameter-free purpose, as it directly provides a *known* upper bound on $B_{\star}$.
> > Specifically, how does the parameter-free algorithm work without this extra assumption? which is for example the case if the number of episodes $K$ is unknown in advance, or if we just run one episode of SSP (i.e., $K=1$), where we would still expect to have a valid regret bound.
> > Could the authors please clarify this point (which I noticed after my initial review)?

---

> > > ### Author Response · Authors · 2021-08-18
> > > **Response to additional comments**
> > >
> > > Response to additional comments:
> > > - **Minor follow-up comment on explicit finite-horizon approximation:** We believe that this is more of an issue of the current analysis instead of a fundamental problem of explicit finite-horizon approximation. We will rephrase accordingly to make this point clear. Thanks again for pointing this out.
> > > - **Shared comment with Reviewer HJ7d on $c_{\min}$/efficiency**: Indeed, when the perturbation trick is applied, $1/c_{\min}$ becomes $K$-dependent and worsens the time complexity of our algorithm. However, note that the EB-SSP algorithm also suffers the same drawback with an even worse dependence $1/c_{\min}^2$, and our comparison discussed in the paragraph "Comparison with EB-SSP" after Theorem 5 is still valid. In fact, this seems to be a shared limitation of all algorithms that learn a stationary policy. On the other hand, the algorithm of [Cohen et al., 2021] can replace the $B_{\star}/c_{\min}$ dependence by $T_{\star}$ (hitting time of the optimal policy), which gives a better time complexity with no poly dependence on $K$ when $c_{\min}=0$, but this only works when $T_{\star}$ is known (otherwise the same issue exists). We will further clarify these points in the revision.
> > > - **New concern/question on the parameter-free algorithm for LCB-ADVANTAGE-SSP:** This is a great point. In the current version we assume $K$ is large enough for simplicity. However, this can be fixed without changing the bound (up to log factors). First, note that our current "parameter free'" algorithm achieves the desired bound as long as we have a valid upper bound on $B_{\star}$, and the regret bound scales only logarithmically w.r.t this upper bound. Therefore, when we don't have an upper bound on $B_{\star}$, we can do the following doubling trick similar to the parameter-free version of our model-based algorithm: divide the learning process into epochs indexed by $\phi$. In epoch $\phi$ we maintain an estimated upper bound $b_{\phi}$ on $B_{\star}$ starting with $b_1=K$, and run our current version of "parameter-free" LCB-ADVANTAGE-SSP with this upper bound $b_{\phi}$. Whenever: (1) $B>b_{\phi}$, or (2) $C_{\phi} > b_{\phi}K + x\left( b_{\phi}\sqrt{SAK} + \frac{b_{\phi}^5S^2A}{c_{\min}^4} \right)$ where $x$ is whatever hidden in the $\tilde{O}$ notation in Theorem 3, we start a new epoch $\phi+1$ and double the estimate: $b_{\phi+1}=2b_{\phi}$. The analysis goes as follows: if $B_{\star}\leq K$, then clearly we will only have one epoch and achieve the desired regret bound. If $B_{\star}> K$, define $\phi^{\star}=\inf_{\phi}( b_{\phi}\geq B_{\star})$. Then, for all $\phi<\phi^{\star}$, we have $C_{\phi}\leq b_{\phi}K + \widetilde{O}\left( b_{\phi}\sqrt{SAK} + \frac{b_{\phi}^5S^2A}{c_{\min}^4} \right)=\widetilde{O}(\frac{B_{\star}^5S^2A}{c_{\min}^4})$, and in epoch $\phi^{\star}$, we achieve the desired regret bound since the upper bound is valid. Summing up these bounds and using the fact $\phi^{\star}=O(\log_2B_{\star})$, we get the desired regret bound $\widetilde{O}(B_{\star}\sqrt{SAK} + \frac{B_{\star}^5S^2A}{c_{\min}^4})$ (the lower order term has an extra factor of $\log_2B_{\star}$). If further $K$ is unknown, we can simply perform the standard doubling trick on $K$ and get a regret bound with an extra $\log_2K$ factor. We will incorporate these discussions in the revision. Thanks again for pointing this out.

---

> > > > ### Comment · Reviewer_WgWg · 2021-08-19
> > > > **Response**
> > > >
> > > > Thanks for these clarifications, which will be helpful for future readers once incorporated in the revised version. I naturally maintain my positive evaluation of the paper.

---

### Official Review · Reviewer_U8Kt · 2021-07-24

**Rating:** 7
**Confidence:** 4

**Summary:**

This paper presents algorithms for the stochastic shortest path (SSP) problem with unknown stochastic transitions but “known and deterministic” costs. The first algorithm, LCB-Advantage-SSP, is the first model-free for SSP with provable regret. When the minimal cost $c_{\min}$ is non-zero, the algorithm achieves the minimax-optimal regret. Notably, LCB-Advantage-SSP has a lower computational complexity compared to existing model-based algorithms for SSP. The second algorithm, SVI-SSP, is a model-based algorithm achieving the minimax-optimal regret even when $c_{\min} = 0$, while requiring less computational complexity in comparison to existing algorithms.

**Limitations And Societal Impact:**

Please pay attention to my detailed comments above.

**Main Review:**

The paper investigates a concrete and interesting RL problem, which received increasing attention during the last two years, though early literature outside of RL, for the fully specified case, abound and date back to decades ago. From a practical standpoint, I believe SSP is at least as significant as the fixed-horizon episodic RL.

Existing literature already present some minimax-optimal algorithms for SSP, but are all model-based, to my best knowledge, which may incur rather high computational costs. Thus, by presenting the first model-free algorithm for SSP, which happens to be minimax-optimal under $c_{\min}>0$, the paper provides a nice addition to the literature. The second algorithm also makes a further step in advancing state-of-the-art, as it requires less computation than state-of-the-art.

The paper is written and organized very well, and almost, typo-free. It admits a clear and precise presentation and is overall a nice read.

Despite these strengths, I have some concerns detailed below:

- While assuming prior knowledge on the cost/reward function has become sort of standard practice in theoretical RL, relaxing this is not always easy or straightforward, notably for model-free algorithms. I am wondering for unknown and “stochastic” costs, what the right way to initialize $Q$ in Algorithm 1 would be. I’m afraid that we may need to know $c_{\min}$ (or some LB on that), to maintain optimism. I may ask the authors to explain this. This may entail further clarification (or refinement) of the remark in the footnote of page 3. If true, the claims in the abstract and introduction should be refined accordingly and explicitly.

- I found it really strange that poly-log terms were made hidden in the $\widetilde O$ throughout (even in the appendix). It is fine to use $\widetilde O$ to arrive at a concise regret bound hiding both numerical constants and poly-log terms. But doing the same practice may not be a good idea for the appendix. At least for the leading-term in the regret bound, it makes sense to report all the constants and logarithmic terms involved. As reviewer, I would have liked to check such dependencies, but it seems that you left it as an exercise to me (and future readers). This is sadly the case for all the bounds and I am afraid that this could partially violate 2(b) in the Checklist.

- As there already exist a couple of minimax-optimal algorithms for SSP, it might render necessary to compare their corresponding regret bounds in terms of the second-order terms. In particular, those terms in the case of LCB-Advantage-SSP seem to have a prohibitive dependence on $c_{\min}$ and $B_\star$, which could be refined via a sharper analysis. This could be of importance from a statistical efficiency perspective, as in the low-K regime, one may not easily ignore the second-order terms.

This is a well-written paper tackling an interesting problem. But I believe these limitations need to be addressed before acceptance. I would be happy to read the authors responses to my raised comments, and upon convincing responses/promises I would consider increasing my score.

Some minor comments follow.

Line 474: $\tilde s_0$ ---- > $\tilde s_{\mathrm{init}}$

Line 879: Taking … initiate ---- > … initiates

Line 140: after-state ---- > the term “next-state” seems to be more common in the literature.

Line 271: two parameter ---- > two parameters


**Time Spent Reviewing:**

8 hours

---

> ### Author Response · Authors · 2021-08-09
> **Response to reviewer U8Kt**
>
> Thanks for your valuable feedback. We will incorporate all minor comments on the writings. Below, we provide detailed response to your major concerns:
> - **Generalization to unknown stochastic costs:** this is a good point, and we will clarify in the revision. In a word, this can indeed be addressed similarly to previous works, even for the model-free algorithm. Specifically, for both of our algorithms, we initialize $Q(s, a)$ to $0$ for all $(s, a)$, and maintain an empirical cost estimator $\widehat{c}(s, a)=\frac{C(s, a)}{n(s, a)}$, where $C(s, a)$ is the accumulated costs suffered at $(s, a)$ and $n(s, a)$ is the total number of visits to $(s, a)$. Then, we replace $c(s, a)$ in the update rule by $\widehat{c}(s, a)$ and add a term of order $\widetilde{O}(\sqrt{\frac{\widehat{c}(s, a)}{n(s, a)}})$ to the bonus $b$ and $b'$. With these modifications, we can obtain the same bounds for both algorithms following the analysis of the cost estimator in [Cohen et at., 2021] (where the definition of $c_{\min}$ now becomes the lower bound of the support of all costs).
> Indeed, the above still requires setting parameters such as $H$ in terms of the unknown quantity $c_{\min}$. However, we can deal with this issue in the same way as we deal with the case $c_{\min}=0$ (discussed in Lines 120-123). Specifically, for a parameter $\epsilon$, we clip all observed costs to $\epsilon$ if they are below $\epsilon$, and then we replace all $c_{\min}$ with $\epsilon$ in our algorithm. For the model-based algorithm, we set $\epsilon = \frac{1}{K}$, and the leading term of the regret bound remains the same thanks to the logarithmic dependence on $\frac{1}{c_{\min}}$ of this algorithm. For the model-free algorithm, however, we need to take $\epsilon=K^{-1/5}$ and obtain a final regret bound of order $K^{4/5}$ as we discuss in Lines 277-279.
>
>
>
> - **Hidden poly-log terms in $\widetilde{O}$:** we decided to hide most of these terms since otherwise the computation quickly becomes tedious and unreadable unfortunately. We can consider spelling out some of these terms if that does not affect the readability, but we do want to emphasize that hiding poly-log terms in $\widetilde{O}$ is mathematically sound and a common practice.
>
>
>
> - **Comparing lower order terms:** We briefly discuss the lower order term after Theorem 5 in the main text. To give a more thorough comparison, we plan to include the following table in our revision ($T_{\star}=\max_sT^{\pi^{\star}}(s)$):
>
>
>
>
>
> |Algorithm| Regret Bound |
> | -- | -- |
> | UC-SSP [Tarbouriech et al., 2020] | $\widetilde{O}( DS\sqrt{\frac{D}{c_{\min}}AK} + S^2AD ^2 )$ |
> | Bernstein-SSP [Cohen et al., 2020] | $\widetilde{O}(B_{\star} S\sqrt{AK} + \sqrt{\frac{B_{\star}^3S^2A^2}{c_{\min}}})$ |
> | ULCVI [Cohen et al., 2021] | $\widetilde{O}(B_{\star}\sqrt{SAK} + T_{\star}^4S^2A)$ |
> | EB-SSP [Tarbouriech et al., 2021] | $\widetilde{O}(B_{\star}\sqrt{SAK} + B_{\star} S^2A)$ |
> | SVI-SSP | $\widetilde{O}(B_{\star}\sqrt{SAK}+B_{\star}S^2A)$ |
> | LCB-ADVANTAGE-SSP | $\widetilde{O}(B_{\star}\sqrt{SAK}+B_{\star}^5S^2A/c_{\min}^4)$ |
>
> Note that the lower order term of SVI-SSP matches that of EB-SSP. LCB-ADVANTAGE-SSP indeed has a large lower order term of order $\widetilde{O}(\frac{1}{c_{\min}^4})$, but this actually almost matches that of ULCVI when $T_{\star}$ is unknown, since in that case their algorithm is run with $T_{\star}$ replaced by its upper bound $\frac{B_{\star}}{c_{\min}}$.

---

> > ### Comment · Reviewer_U8Kt · 2021-09-09
> > **Response to Rebuttal**
> >
> > Thanks for the rebuttal.
> >
> > I have read all the reviews and the rebuttal. The rebuttal addresses my concern. In particular:
> >
> > - I find the table comparing the lower order terms quite informative.
> >
> > - As far as I see, the issue of unknown stochastic costs were also raised by two other reviewers, so inclusion of an appropriate discussion in this regard in the revision would be great.
> >
> > - I am completely familiar with the standard practice of hiding poly-log terms in $\tilde O(.)$ to improve readability, specially in the main text. And I admittedly use it to a great extent. On the other hand, I tend to think that these could be more elaborated in the appendix. One supporting argument could be that some readers might be looking for such precise representations for, e.g., comparison purposes (at least for the leading terms). Let me stress that hiding a standard logarithmic term, e.g., $O(\sqrt{\log(K/\delta)})$ might be completely different than hiding something like $O(\log^6(K/\delta))$. So more spelled out leading terms, as suggested in the rebuttal, would be perfect.
> >
> > Finally, incorporating the comments of other reviewers improves this already well-executed paper. Assuming that promised changes will be made in the revision, I increase my score to 7.

---

### Decision · Program_Chairs · 2021-09-27

**Decision:**

Accept (Poster)

**Comment:**

The paper presents a framework for the analysis of tabular SSP problems which is used to prove regret bound for both model-based and model-free algorithms. The model-based algorithm matches the best-known rate while allowing sparse updates. More interesting, the paper introduces the first model-free algorithm for this setting. The paper is solid, novel and interesting. It is a valuable contribution to the conference.

While the reviewers all agree on the value of this paper, they provided relevant suggestions. Please take them into consideration in the revised version.